# Statistical theory of probabilistic hazard maps: a probability distribution for the hazard boundary location

David M. Hyman[1,2], Andrea Bevilacqua[2,3,4], and Marcus I. Bursik[2]

[1]Collaborative Institute for Meteorological Satellite Studies (CIMSS), University of Wisconsin - Madison, WI
[2]Department of Geological Sciences, University at Buffalo, NY
[3]Computational Data Science and Eng., University at Buffalo, NY
[4]Istituto Nazionale di Geofisica e Vulcanologia (INGV), Sezione di Pisa, Pisa, Italy

**Correspondence:** David M. Hyman (dave.hyman@ssec.wisc.edu)

**Abstract.** The study of volcanic flow hazards in a probabilistic framework centers around systematic experimental numerical modelling of the hazardous phenomenon and the subsequent generation and interpretation of a probabilistic hazard map (PHM). For a given volcanic flow (e.g., lava flow, lahar, pyroclastic flow, ash cloud, etc.), the PHM is typically interpreted as the point-wise probability of inundation by flow material.

In the current work, we present new methods for calculating spatial representations of the mean, standard deviation, median, and modal locations of the hazard's boundary as ensembles of many deterministic runs of a physical model. By formalizing its generation and properties, we show that a PHM may be used to construct these statistical measures of the hazard boundary which have been unrecognized in previous probabilistic hazard analyses. Our formalism shows that a typical PHM for a volcanic flow not only gives the point-wise inundation probability, but also represents a set of cumulative distribution functions for the location of the inundation boundary with a corresponding set of probability density functions. These distributions run over curves of steepest probability gradient ascent on the PHM. Consequently, 2D space curves can be constructed on the map which represent the mean, median and modal locations of the likely inundation boundary. These curves give well-defined answers to the question of the likely boundary location of the area impacted by the hazard. Additionally, methods of calculation for higher moments including the standard deviation are presented, which take the form of map regions surrounding the mean boundary location. These measures of central tendency and variance add significant value to spatial probabilistic hazard analyses, giving a new statistical description of the probability distributions underlying PHMs.

The theory presented here may be used to aid construction of improved hazard maps, which could prove useful for planning and emergency management purposes. This formalism also allows for application to simplified processes describable by analytic solutions. In that context, the connection between the PHM, its moments, and the underlying parameter variation is explicit, allowing for better source parameter estimation from natural data, yielding insights about natural controls on those parameters.

# 1 Introduction

The probabilistic study of volcanic hazards is part of an emerging paradigm in physical volcanology, one that openly admits and seeks to calculate and characterize the uncertainty inherent to physical modelling of hazardous volcanic processes (Bursik et al, 2012; Connor et al., 2015; Bevilacqua, 2016; Sandri et al., 2016). Critically, this approach is founded on the notion that physical parameters related to these processes can be represented by a probability distribution. The distribution is propagated by a physical model to yield a space of hazard distributions which put probabilistic constraints on quantities that would be fixed in a deterministic, scenario-based modelling framework. In practice this has typically been accomplished by accumulating the statistics of the processes through deterministic modelling over an ensemble of model inputs (Neri et al., 2015; Biass et al., 2016; Tierz et al., 2016; Bevilacqua et al., 2017; Gallant et al., 2018; Patra et al., 2018b). Polynomial chaos approximations and Gaussian model surrogates can provide fast algorithms to obtains these statistics (Bursik et al, 2012; Madankan et al., 2012; Stefanescu et al., 2012; Spiller et al., 2014; Bayarri et al., 2015; Tierz et al., 2018). Owing to the nature of the studied processes, the results are typically displayed in map form and contain a large amount of spatial information, the interpretation of which has become a source of extensive study (Calder et al., 2015; Thompson et al., 2015). Because the information in these maps is potentially of great interest to a wide community of stakeholders including governments, planners, emergency managers, local residents, and others spanning a wide range of familiarity with probabilistic reasoning, finding the best approach to condensing these data-rich maps is non-trivial. The raw probabilities are considered less useful to many end users despite their significant information content (Thompson et al., 2015). Often the information in these maps is presented qualitatively as "high", "medium", and "low" hazard or simply "high" and "low" hazard (Calder et al., 2015; Wagner et al., 2015). The reduction of a complex probability distribution to a relative categorization as in a high - low hazard scheme generally involves significant nuance in blending scientific information, decision - making, and administrative considerations. In the scientific component of the analysis, such a categorization could be based on a probability threshold, in many cases a certain percentile of the distribution, which corresponds to a probability contour on the map. While this is the most obvious method it is just one of many options for declaring this categorical boundary.

At present, a probabilistic hazard map (PHM) for a given hazard is easily constructed from numerical modelling; however, its most commonly used statistical product in hazard communication is just the set of probability contours. Consequently if a binary volcanic hazard map were to be constructed from a PHM showing the region of no likely impact versus the region of likely impact, the analyst would have to decide where to draw the line: possibly at the $5\%$ probability contour (a safe, conservative choice), the median contour ($50\%$) (a good estimate for the most likely impact - no impact boundary), or some other choice of percentile. Often the choice of the contour level is based on agreements or official protocols made by civil protection authorities. Such potentially high-consequence decisions would benefit from additional statistical analyses which generate other tools for this task. For example, it may be useful to know the average geographical limits of a particular hazard in addition to the point-wise probability of hazard impact. For any sample data set or theoretical probability distribution, the mean and variance as well as one or more modal values are calculable in addition to the median and other percentiles. An important deficiency in the analysis of the PHM is that previously, scientific estimates of the likely hazard boundary (a single

curve on the map) have lacked the formal framework required for unique determination of statistical estimates such as the mean, variance, and higher moments of the distribution of hazard boundary locations.

For any given volcanic hazard analysis which maps probabilities of a given event, the probability values may represent a wide array of concepts in probability theory including cumulative probabilities, probability densities, conditional probabilities, and others. Here, the point-wise probability is interpreted as the relative frequency of hazard impact as predicted by repeated simulation using a deterministic physical model or ensemble of models equipped with an ensemble of inputs spanning the realistic range of underlying uncertain parameters. In the following theory and analysis, we only consider spatial probability distributions which map the probability of a given hazard impacting a given location where there is at least one point in space with a modelled probability of unity. Although this excludes the study of many possible PHMs, it covers a sufficiently wide range of models which are regular or regularized and span a realistically limited subspace of the general parameter space, that is, they depend continuously on the inputs or in this case, maintain some level of spatial correlation across a realistically narrow, continuous range of inputs. In general, the methods detailed here are agnostic with respect to the type of uncertainty which is modelled. For any probabilistic geophysical model, there exist epistemic and aleatory uncertainties. The division between aleatory uncertainty and epistemic uncertainty corresponds to the distinction between intrinsic randomness of the system and the additional uncertainty that affects its representation, the latter originating from incomplete information about the underlying physical processes. Clearly PHMs constructed purely from aleatory uncertainties in the inputs have the most concrete meanings. However, PHMs constructed from a mix of aleatory and epistemic uncertainties can be considered as a single member of a time-dependent sequence of such assessments, each of which captures the types of uncertainty encoded in the generation of the PHM that is understood at that time. In general, as epistemic uncertainty decreases over time, the later assessments will become more accurate. As long as the uncertainties can be described as a probability space, these analyses hold up to current geophysical knowledge. Although varying the parameters in a single model gives the most concrete meaning, these methods could be applied to PHMs constructed from an ensemble of models provided that the ensemble shows regular behavior. Here, we focus on analyses of single hazards, although the boundary-finding methods could be justified to analyze PHMs constructed for multiple hazards.

Throughout this work, we realize these methods with examples using geophysical mass flow (GMF) models of inundation hazards. This is done primarily because the flow boundary is easily definable and conceptualized in these models, giving a consistent intuition to the complicated mathematical quantities discussed here. However, these concepts extend to many other types of hazard models where a particular threshold is of interest in defining the region impacted by a hazard in volcanology and other fields including concentration thresholds in volcanic clouds (Bursik et al, 2012), thickness thresholds in tephra fallout (Sandri et al., 2016), ballistic ejecta impact count thresholds (Biass et al., 2016), pyroclastic density current invasion (Bevilacqua et al., 2017), lava flow inundation (Gallant et al., 2018), tsunami runup thresholds (Geist and Parsons, 2006; Grezio et al., 2017), ground shaking thresholds in seismic hazards (Kvaerna et al., 1999), hurricane wind speed thresholds (Splitt et al., 2014), and many others. All of these examples can be cast in the same mathematical formulation that we detail below. A full evaluation of suitable and unsuitable types of probabilistic hazard assessments is beyond the scope of this work.

The work of Stefanescu et al. (2012) provided a rigorous framework for calculating the PHM from a Gaussian emulator of GMF simulations. Within that framework, the PHM is the point-wise mean probability of hazard impact and the uncertainty bounds for that probability are well-defined. Although this method is robust, it has not been applied in many hazard studies. Furthermore, as with most calculations of PHMs, the interval $0 < Prob(Hazard\,Impact) < 1$ occupies a very large region of space. As in the above hypothetical binary hazard mapping scenario, the question remains - From the perspective of a probabilistic volcanic hazard analysis, where is the likely boundary of the impacted area?

The goal of the present work is to extend the probability theory of PHMs to construct statistically meaningful answers to this question. To this end, we give an explicit mathematical definition of the PHM, defining explicitly the connection between the cumulative exceedence probabilities and probability densities in a PHM and estimates of the likely hazard boundary. More specifically, we calculate spatial representations of the PHM's moments including the mean and variance of the hazard boundary location as well as its modality. We stipulate that the methods detailed here apply only to the scientific analysis component of any hazard mapping project and must be considered as just one piece of the hazard map-making process. This effort combines mathematical tools and concepts from probability theory, the study of dynamical systems, and differential geometry.

## 2    Hazard Map Distribution Theory

Solutions to flow models which involve depth-integrated partial differential equations (e.g., Saint-Venant-type shallow water equations) typically include solving for the distribution of flow thicknesses over position and parameter space:

$$h = h(\boldsymbol{x}; \boldsymbol{\beta}) \tag{1}$$

where $\boldsymbol{x}$ is the position vector and $\boldsymbol{\beta} \in \mathcal{B}$ is an n-vector in parameter space $\mathcal{B}$, that is, if there are $n$ input parameters to a model, $\mathcal{B} \subset \mathbb{R}^n$. The parameter space $\mathcal{B}$ is taken as a probability space; however, we discuss it here only in a simplified sense without reference to sigma-algebras. For the present purposes, we note only that $\mathcal{B}$ contains all values of the input parameters that may be considered reasonable or feasible for the application. The notation $h(\boldsymbol{x}; \boldsymbol{\beta})$ has been used to indicate that the model output is a function of position $\boldsymbol{x}$ that depends upon the input parameters in $\boldsymbol{\beta}$. For any choice of model, the vector $\boldsymbol{\beta}$ is simply a collection of the input parameters to the model. From the solutions $h(\boldsymbol{x}; \boldsymbol{\beta})$, an indicator function can be constructed to denote the region in space, $\Omega(\boldsymbol{\beta})$, where the solution space has been inundated:

$$\mathbf{1}_{\Omega(\boldsymbol{\beta})}(\boldsymbol{x}) := \begin{cases} 1 & \{\boldsymbol{x} \mid h(\boldsymbol{x}; \boldsymbol{\beta}) > h_0\} \\ 0 & \{\boldsymbol{x} \mid h(\boldsymbol{x}; \boldsymbol{\beta}) \leq h_0\} \end{cases} \tag{2}$$

where $h_0$ is the inundation thickness threshold of interest. For many GMFs including lahars, lava flows, and pyroclastic density currents, the threshold of interest is $h_0 = 0$ since any amount of inundation is a significant hazard; however, in the typical practice of PHM construction from numerical modelling, a non-zero thickness tolerance is typically set due to inaccuracies in the solver near free boundaries ($h(\boldsymbol{x}) = 0$) or other considerations (e.g., Patra et al., 2018b). Although doing so is inaccurate

for the flows considered in this work, a sufficiently small threshold captures the essential features of the flow boundary. In assessments of other types of hazards with non-zero thresholds of interest (e.g. tephra fallout required for collapse of a particular roofing material), the true threshold may be used without introducing problems due to these solver inaccuracies.

## 2.1 Probabilistic Hazard Map

5  The construction of any PHM requires the definition of a joint probability distribution of the inputs given by a probability density function (PDF) $f : \mathcal{B} \to \mathbb{R}^+$ which measures the parameter space $\mathcal{B}$. In general, the choice of probability distribution for $\boldsymbol{\beta}$ will represent a combination of aleatory and epistemic uncertainties. Consequently, calculating the PHM amounts to propagating $f(\boldsymbol{\beta})$ from parameter space ($\mathcal{B}$) through a physical model into Euclidean space. The PHM is then calculated by integration through all of parameter space:

$$\phi(\boldsymbol{x}) := \int_{\mathcal{B}} f(\boldsymbol{\beta}) \mathbf{1}_{\Omega(\boldsymbol{\beta})}(\boldsymbol{x}) \, d\boldsymbol{\beta}. \tag{3}$$

In the integral, the indicator function acts as a filter, removing parameter space points from the integral which do not yield solutions exceeding the threshold at a given space point. In spatial subsets where all values of $\boldsymbol{\beta}$ in the support of $f$ give solutions which exceed the threshold, the indicator is identically unity and $\phi = 1$ by the scaling condition for a valid PDF $f(\boldsymbol{\beta})$. Consequently, $0 \leq \phi \leq 1$ globally and measures the probability of finding material in the simulated flow at position $\boldsymbol{x}$.

An alternative interpretation is the following: any given location exists in a binary state, either "impacted" or "not impacted" by the flow, which is represented spatially by an indicator function with the model acting as a mapping from a set of inputs $\boldsymbol{\beta}$ to the indicator $\mathbf{1}_{\Omega(\boldsymbol{\beta})}(\boldsymbol{x})$. Invoking the "law of the unconscious statistician" (LOTUS) (DeGroot and Schervish, 2012), the PHM is then a formula for computing the probability of being in either member of the binary state at each point, since the indicator function (model output) is integrated against the PDF for the model inputs. This second interpretation (using the LOTUS) is

the result of encoding the binary state as a statement of certainty (probability zero or unity) that a given point is inundated in each model output.

If the range of realistic parameter values were confined to a single point $\boldsymbol{\beta}_0$ in parameter space, then the parameter PDF $f$ is the Dirac delta function $\delta(\boldsymbol{\beta} - \boldsymbol{\beta}_0)$, resulting in a PHM which is merely the indicator function: $\phi(\boldsymbol{x}) = \mathbf{1}_{\Omega(\boldsymbol{\beta}_0)}(\boldsymbol{x})$. This is also the result for a PHM constructed for a single deterministic run of the model. This corresponds to the intuition that complete

certainty of input parameters corresponds with complete certainty of the spatial extent of the hazard (the region where $\phi = 1$).

If the parameter space is measured by a uniform distribution (maximal entropy distribution), the probabilistic hazard map (PHM) is constructed by mean-value integration of the indicator function through the parameter space:

$$\phi(\boldsymbol{x}) = \fint_{\mathcal{B}} \mathbf{1}_{\Omega(\boldsymbol{\beta})}(\boldsymbol{x}) \, d\boldsymbol{\beta} \tag{4}$$

where the symbol $\fint$ is the mean value integral. This approach is a good assumption in hazard analysis if there is no *a priori*

knowledge of the uncertain parameters except for reasonable bounds.

In the context of a typical probabilistic hazard modelling effort, the solution space and parameter space are discrete with a uniform distribution in parameter space and these steps are accomplished merely by cell-wise averaging of the indicator func-

tions derived from each model run to generate a discretized 2D hazard map $\phi = \phi_{ij}$. In that context, $\phi_{ij}$ is typically contoured and presented in a hazard assessment as probabilistic zones defined by the specific contours. However, because the underlying physics of these flows is not well constrained, the parameter space must be sampled widely resulting in large probability dispersion in some cases. In the present work we globally analyze the PHM as a type of probability distribution and generate its associated measures of central tendency which form curves in 2D space. To analyze the PHM, we must first note several of its features and state some assumptions critical to the analysis. For a PHM generated from a finite, simply-connected parameter space and physically realistic governing equations:

$(i)$ $\phi(\boldsymbol{x})$ has compact support in a region $\Omega^0 \subset \mathbb{R}^2$, with $\phi = 0$ on $\partial\Omega^0$.

$(ii)$ $0 \leq \phi \leq 1$

$(iii)$ There exists a (possibly unconnected) set $\Omega^1 \subset \Omega^0$ in which $\phi(\boldsymbol{x}) = 1$.

We will further assume that $\phi(\boldsymbol{x})$ is continuous and at least twice-differentiable for $\forall \boldsymbol{x} \in \mathbb{R}^2 \setminus \{\partial\Omega^0 \cup \partial\Omega^1\}$, that is, in the regions away from the $\phi = 0$ and $\phi = 1$ contours. We focus the analysis to follow in regions away from local maxima with $\phi < 1$. This restriction can be mitigated by segmentation of the spatial domain as detailed below.

Formally, $\phi(\boldsymbol{x})$ is a cumulative distribution function (CDF) for the location of the edge of the simulated flow. This is represented schematically in Fig. 1. To see this, consider parametric curves along level set unit normals of $\phi$:

$$\frac{d\boldsymbol{x_n}}{ds} = \hat{\boldsymbol{n}} := \frac{\nabla\phi}{|\nabla\phi|} \tag{5}$$

parameterized by arc length $s$ with $0 \leq s \leq L$ where $\boldsymbol{x_n}(0) \in \partial\Omega^0$ and $\boldsymbol{x_n}(L) \in \partial\Omega^1$ and $L$ is the finite total Euclidean path length of $\boldsymbol{x_n}$. Note that $\hat{\boldsymbol{n}}$ is only calculable where $|\nabla\phi| \neq \boldsymbol{0}$. To ameliorate this, we define $\hat{\boldsymbol{n}}$ piecewise as

$$\hat{\boldsymbol{n}}(\boldsymbol{x}) := \begin{cases} \nabla\phi/|\nabla\phi| & \text{where } |\nabla\phi| \neq \boldsymbol{0} \\ 0 & \text{elsewhere} \end{cases} \tag{6}$$

Along each integral curve, profiles of the PHM are denoted

$$\phi(s) := \phi(\boldsymbol{x_n}(s)) = \mathbb{P}(s_F \leq s) \tag{7}$$

where $\mathbb{P}(s_F \leq s)$ is the cumulative probability that position $\boldsymbol{x_n}(s)$ is farther from $\boldsymbol{x_n}(0)$ than the true inundation front $\boldsymbol{x_n}(s_F)$ is from $\boldsymbol{x_n}(0)$ as measured along the integral curve. For example, where $\phi(s) = 1$, the inundation front is almost surely found somewhere further back on the curve towards $\boldsymbol{x_n}(0)$. Similarly, where $\phi(s) = \frac{1}{2}$, there is a $50\%$ chance of finding the true inundation front between that point ($\boldsymbol{x_n}(s)$) and either of the two endpoints of the integral curve ($\boldsymbol{x_n}(0)$ and $\boldsymbol{x_n}(L)$). Because the integral curves follow the gradient, $\phi(s)$ is non-decreasing, and along with $0 \leq \phi(s) \leq 1$ satisfies the description of a CDF.

If local maxima less than unity are present, then the integral curves that ascend these maxima will not be valid CDFs ($\phi(s)$) because they never reach the value $\phi = 1$. Despite this, as long as the PHM contains a finite number of local maxima less than unity ($\phi(\boldsymbol{x_m}) < 1$ where the index $m$ runs over the $M$ local maxima) then for each local maximum, we may define a

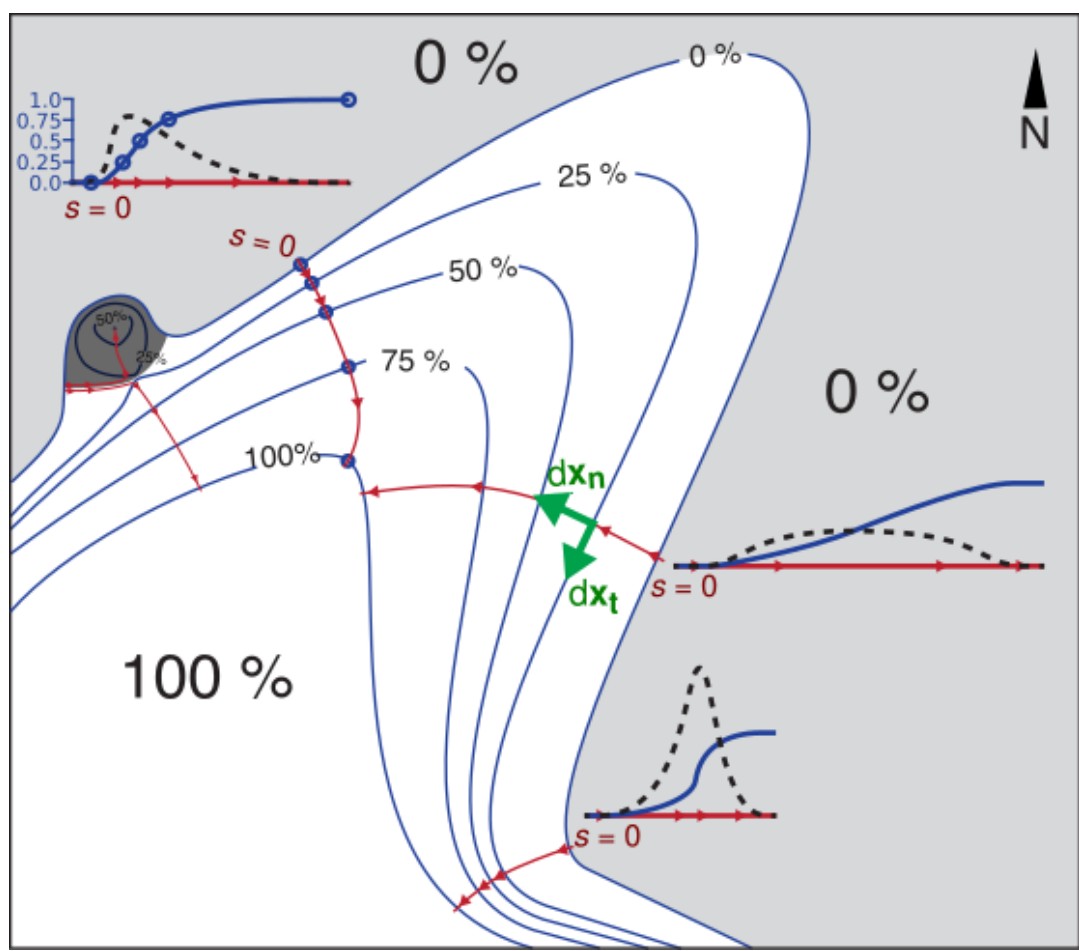

**Figure 1.** Schematic representation of a hypothetical probabilistic hazard map (PHM) where contours (blue) are the percent of deterministic models inundating a given point in space. Steepest ascent integral curves on the PHM ($\boldsymbol{x_n}(s)$) are shown in red. Profiles for the cumulative distribution $\phi(\boldsymbol{x_n}(s))$ (blue) and the probability density $d\phi(\boldsymbol{x_n}(s))/ds$ (black dashes) are represented schematically. The dark grey transparent region is the region of influence for the local maximum ($\mathcal{R}_m$) in which all integral curves terminate at the local maximum. Two integral curves (one inside of $\mathcal{R}_m$, one outside of $\mathcal{R}_m$) show the typical behavior of $\boldsymbol{x_n}(\boldsymbol{s})$ near $\partial\mathcal{R}_m$. The set $\widetilde{\Omega}^0$ is the entire white region, $\Omega^0$ is the region bounded by the zero level set (the white region plus the dark grey region), and $\Omega^1$ is the subset of that region which is marked with 100%.

region $\mathcal{R}_m$ which surrounds $\boldsymbol{x}_m$ and is bounded by two integral curves as well as a segment of $\partial\Omega^0$ and $\partial\Omega^1$. Because local maxima are sinks for gradient-ascending integral curves, all of the integral curves that begin on $\partial\mathcal{R}_m \cap \partial\Omega^0$ will terminate at $\boldsymbol{x}_m$ by definition. The two bounding integral curves are therefore defined extrinsically as those integral curves producing valid CDFs originating closest to the segment of $\partial\Omega^0$ that originates the integral curves terminating at $\boldsymbol{x}_m$. Although these regions are only defined in a weak sense, these regions are definable for practical purposes (Appendix B). If there is a saddle point between a local maximum and $\partial\Omega^1$, then the two integral curves will meet at the saddle point and no part of $\partial\Omega^1$ will make

up the boundary of $\mathcal{R}_m$. In a sense, $\mathcal{R}_m$ can be thought of as the "shadow" of trajectories that are influenced most by the local maximum $\phi(\boldsymbol{x}_m)$ (Fig. 1). We remove these regions by constructing a new set on which the remainder of the analysis is valid:

$$\widetilde{\Omega}^0 = \Omega^0 \setminus \bigcup_{m=1}^{M} \mathcal{R}_m. \tag{8}$$

## 2.2   Probabilistic Hazard Density Map

To find the PDF $d\phi/ds$ associated with this distribution, consider that by the chain rule:

$$\frac{d\phi(s)}{ds} = \left(\nabla\phi \cdot \frac{d\boldsymbol{x}}{ds}\right)\bigg|_{\boldsymbol{x}_{\boldsymbol{n}}(s)} = |\nabla\phi(\boldsymbol{x}_{\boldsymbol{n}}(s))| \tag{9}$$

and consequently,

$$\int_0^L \frac{d\phi}{ds}\,\mathrm{d}s = \int_0^L |\nabla\phi(\boldsymbol{x}_{\boldsymbol{n}}(s))|\,\mathrm{d}s = 1. \tag{10}$$

Although this description satisfies the requirements of a PDF along each individual integral curve, it cannot be applied directly to measure the probability density in two dimensions. To measure this probability density, we generate a normalized PDF over 2D space which we refer to as the probabilistic hazard density map (PHDM):

$$\psi(\boldsymbol{x}) = \frac{1}{Q}|\nabla\phi|, \tag{11}$$

where

$$Q = \int_{\widetilde{\Omega}^0} |\nabla\phi|\,\mathrm{d}A = -\int_{\widetilde{\Omega}^0} \phi\nabla\cdot\hat{\boldsymbol{n}}\,\mathrm{d}A \tag{12}$$

which is finite and positive in the sense of distributions. With this normalization, the integral over $\widetilde{\Omega}^0$ is unity, giving $\psi(\boldsymbol{x})$ the properties of a PDF for the inundation front location. Consequently, the value $\psi(\boldsymbol{x})\,\mathrm{d}A$ represents the probability of finding the inundation edge in an increment of area $\mathrm{d}A$. As noted, this distribution is only valid in $\widetilde{\Omega}^0$: the region of space where local maxima in the PHM do not influence the integral curves. A proof which extends Eqn. 10 to this normalization of the PHDM is given in Appendix A.

In the notional example given above for a PHM generated by a parameter space Dirac delta, given a PHM that is identically the indicator function, the PHDM is defined only in the sense of distributions because of the apparently infinite gradient at $\Omega(\boldsymbol{\beta}_0)$. However, because derivatives can be defined in the sense of distributions, the PHDM corresponding to certain input parameters is $\psi(\boldsymbol{x}) = \delta(\boldsymbol{x} - \boldsymbol{x}^*)$ where $\boldsymbol{x}^* \in \partial\Omega(\boldsymbol{\beta}_0)$ form the locus of points representing the certain boundary location. In this case, $\Omega^0 = \Omega(\boldsymbol{\beta}_0)$. This scenario corresponds to the propagation of complete certainty in parameter space to complete certainty of the flow boundary location in Euclidean space.

## 2.3 Measures of Central Tendency and Variance

Using these definitions, three measures of central tendency (mode, mean, median) can be generated for each integral curve, generating a locus of points (curve) for each measure parameterized by a new parameter ($l$) which increases along the PHM contours according to:

$$\frac{d\boldsymbol{x_t}}{dl} = \mathbf{R}\hat{n},$$ (13)

where $\mathbf{R} = \begin{pmatrix} 0 & -1 \\ 1 & 0 \end{pmatrix}$ is a rotation matrix and $\boldsymbol{x_t}(l)$ are integral curves everywhere tangent to the PHM contours (schematized in Fig. 1).

By defining a density function for the inundation edge probability distribution as in Eqn. 11, a notional "modal" set can be constructed by connecting the maximal value of $\psi$ on each integral curve $\boldsymbol{x_n}$. The modal set (parameterized by $l$) is

$$\boldsymbol{x}_{\mathcal{M}}(l) := \{\boldsymbol{x} \in \mathbb{R}^2 \mid \psi(\boldsymbol{x}) = \max_{s \in [0,L]} \psi(\boldsymbol{x_n}(s;l))\}.$$ (14)

For the inundation hazard map, the curve $\boldsymbol{x}_{\mathcal{M}}(l)$ represents the maximum likelihood curve along which to find the inundation front. Although each distribution may not be unimodal and individual distributions may have unimodal and multimodal regions owing to the complexity of the underlying physics of the problem, this estimate gives a starting point for understanding the central tendency of the distribution.

Furthermore, we may find the mean location of the inundation by considering the first moment of each univariate PDF $\frac{d\phi}{ds}(s;l)$. For each integral curve $\boldsymbol{x_n}(s;l)$ connecting a point on $\partial\Omega^0$ to a point on $\partial\Omega^1$, the first moment (mean) of $d\phi/ds$ may be written:

$$\mu(l) = \int_0^L s\frac{d\phi}{ds}(s;l)\,\mathrm{d}s.$$ (15)

We compute this integral by parts, but make a critical substitution before doing so. Note that for a new function $1 - \phi$, differentiating yields $\mathrm{d}\phi = -\mathrm{d}(1 - \phi)$. We may substitute this into Eqn. 15 and then integrate by parts:

$$\mu(l) = -s\big[1 - \phi(s;l)\big]\Big|_{s=0}^{s=L} + \int_0^L 1 - \phi(s;l)\,\mathrm{d}s.$$ (16)

Because $\phi(L) = 1$, the endpoint-evaluated terms in the limit will vanish. This allows for a simpler representation for the first moment:

$$\mu(l) = \int_0^L 1 - \phi(\boldsymbol{x_n}(s;l))\,\mathrm{d}s.$$ (17)

This formula gives the expected value of the arc length parameter, which can be substituted to give an expected boundary location. Once $\mu(l)$ is calculated for every integral curve $\boldsymbol{x_n}(s;l)$, we may connect these points to form a curve defined by:

$$\boldsymbol{x}_{\mu}(l) := \Big\{\boldsymbol{x} \in \mathbb{R}^2 \mid \boldsymbol{x} = \boldsymbol{x_n}\big(\mu(l)\big)\Big\}.$$ (18)

Finally, we may define a curve representing the median value of the probability distribution for inundation front location. This is simply the 50% probability contour:

$$\boldsymbol{x}_{med}(l) := \left\{ \boldsymbol{x} \in \mathbb{R}^2 \mid \phi(\boldsymbol{x}) = \frac{1}{2} \right\} \tag{19}$$

which is a level set of the hazard map. The three curves $\boldsymbol{x}_{\mathcal{M}}$, $\boldsymbol{x}_\mu$, and $\boldsymbol{x}_{med}$ represent different measures of the central tendency of the hazard map probability distribution.

Additionally, a measure of the variance may also be constructed for each PDF $\frac{d\phi}{ds}(s;l)$, although this requires integration of the density function over $s$:

$$\sigma^2(l) = \int_0^L s^2 \frac{d\phi}{ds}(s;l) \, ds - \mu(l)^2. \tag{20}$$

The standard deviation ($\sigma$) defines a region around the mean value in that for each integral curve, the points $s = \mu(l) \pm \sigma(l)$ map to points in space along the curves $\boldsymbol{x}_{\mu \pm \sigma}(l)$, forming a region about $\boldsymbol{x}_\mu(l)$ between $\boldsymbol{x}_{\mu+\sigma}(l)$ and $\boldsymbol{x}_{\mu-\sigma}(l)$. Higher moments of the distribution many be calculated similarly along each curve and subsequently connected.

## 3   Results

To highlight the features of the new theory, we present three examples of its application in order of increasing complexity and realism.

### 3.1   Example 1: Analytic model of time-dependent solidification of a viscous gravity current

A simple application of this theory is to a flow which can be calculated analytically. Here, we show the application of the above theory to the problem of time-dependent solidification of a viscous gravity current. This model used by Quick et al. (2017) is a modification of the famous model due to Huppert (1982) of the axisymmetric (radial) spread of viscous gravity currents, allowing for time dependent viscosity where the height of the current is governed by:

$$\partial_t h - \frac{g}{\nu(t)} \frac{1}{r} \partial_r \left[ r h^3 \, \partial_r h \right] = 0 \tag{21}$$

where $g$ is gravitational acceleration. This model incorporates a simple model of time-dependent kinematic viscosity increase representing the solidification of the current,

$$\nu(t) = \nu_0 e^{t/\Gamma}, \tag{22}$$

where $\nu_0$ is the kinematic viscosity ($\nu$) at time $t = 0$ and $\Gamma$ is the time it takes for the viscosity to increase by a factor of $e$. The solution for the release of a constant volume $V_0$ with initial radius $r_0$ may be stated as:

$$h(r,t) = \frac{4V_0}{3\pi r_0^2} \frac{1}{(1 + \Theta(t)/\tau)^{1/4}} \left[ 1 - \frac{r^2}{r_0^2} \frac{1}{(1 + \Theta(t)/\tau)^{1/4}} \right]^{1/3} \tag{23}$$

where

$$\Theta(t) = \Gamma(1 - e^{-t/\Gamma}) \tag{24}$$

and

$$\tau = \left(\frac{3}{4}\right)^5 \left(\frac{\pi}{V_0}\right)^3 \frac{\nu_0 r_0^8}{g} \tag{25}$$

Notably, $\Theta \to \Gamma$ as $t \to \infty$, implying that eventually the current will solidify and grow to a maximum size. Consequently, when the flow dynamics have ceased, the radius of the edge of the current ($r_N$), where the current thickness vanishes, will have reached a maximum at

$$r_N = r_0(1 + \Gamma/\tau)^{1/8}. \tag{26}$$

To use this simple model to illustrate the probability distribution properties above, consider the case where all parameters

are known precisely except for the viscosity freezing time coefficient $\Gamma$ which can be constrained within a range: $\Gamma_{min} < \Gamma < \Gamma_{max}$. Because Eqns. 23 and 26 could be nondimensionalized so as to be rewritten in terms of one fundamental parameter $\eta := \Gamma/\tau$, which is proportional to $\Gamma$, this single random parameter choice captures the underlying uncertainty in this example. Under the assumption of no prior knowledge of the likelihood of any particular value of $\Gamma$, a uniform distribution may be used to construct an inundation indicator function for this model:

$$\mathbf{1}_{\Omega(\Gamma)}(r) = \begin{cases} 1 & r < r_N(\Gamma) \\ 0 & r \geq r_N(\Gamma). \end{cases} \tag{27}$$

To generate a PHM from this indicator function as above, the indicator function must be integrated through the parameter space. The only nonconstant segment of this integral is on the interval $r_N(\Gamma_{min}) < r < r_N(\Gamma_{max})$ where

$$\phi(r) = \frac{1}{\Delta\Gamma} \int\limits_{\Gamma_N(r)}^{\Gamma_{max}} d\Gamma \tag{28}$$

with $\Gamma_N(r) = \tau[(r/r_0)^8 - 1]$ and $\Delta\Gamma = \Gamma_{max} - \Gamma_{min}$. This results in the PHM (Fig. 2a,c):

$$\phi(r) = \begin{cases} 1 & r \leq r_N(\Gamma_{min}) \\ \frac{\Gamma_{max} + \tau - \tau(r/r_0)^8}{\Gamma_{max} - \Gamma_{min}} & r_N(\Gamma_{min}) < r < r_N(\Gamma_{max}) \\ 0 & r \geq r_N(\Gamma_{max}). \end{cases} \tag{29}$$

From the PHM, the PHDM (Fig. 2b,c) can be calculated as

$$\psi(r) = \frac{1}{Q}|\nabla\phi| = \begin{cases} \frac{9}{2\pi} \frac{r^7}{r_N(\Gamma_{max})^9 - r_N(\Gamma_{min})^9} & r_N(\Gamma_{min}) < r < r_N(\Gamma_{max}) \\ 0 & \text{otherwise}, \end{cases} \tag{30}$$

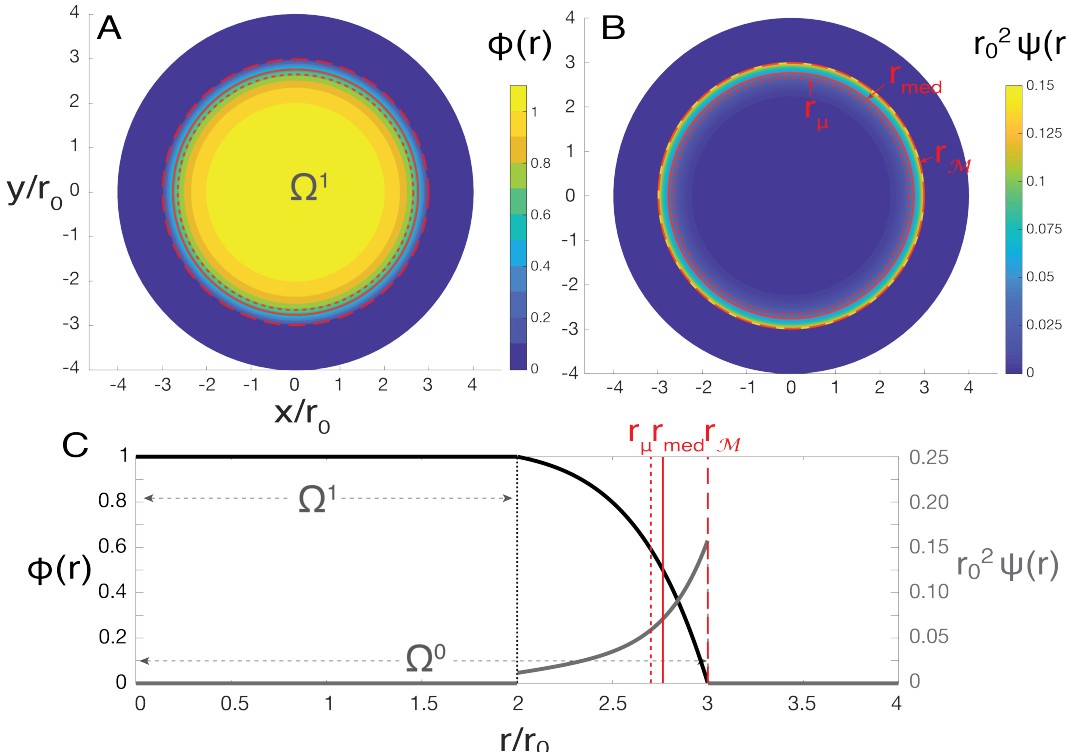

**Figure 2.** a) PHM and B) PHDM examples for the freezing viscous gravity current model under a uniform sampling of the freezing rate parameter $\Gamma$. c) 1D profile of the PHM and PHDM with the distribution centers indicated. This example was calculated with $r_N(\Gamma_{min})/r_0 = 2$ and $r_N(\Gamma_{max})/r_0 = 3$ or $\Gamma_{min}/\tau = 255$ and $\Gamma_{max}/\tau = 6560$.

and the integral curves as

$$r(s) = r_N(\Gamma_{max}) - s \tag{31}$$

with $0 \leq s \leq L$ and $L = r_N(\Gamma_{max}) - r_N(\Gamma_{min})$.

These relationships lead naturally to the estimates of the center of the distribution (Fig. 2) as defined above:

$$5 \quad \begin{cases} r_{\mathcal{M}} = r_N(\Gamma_{max}) & (32a) \\[2mm] r_{med} = r_0 \left[ 1 + \dfrac{\Gamma_{max} + \Gamma_{min}}{2\tau} \right]^{1/8} & (32b) \\[2mm] r_\mu = r_N(\Gamma_{max}) - \mu & (32c) \end{cases}$$

and the standard deviation:

$$r_{\mu \pm \sigma} = r_N(\Gamma_{max}) - \mu \mp \sigma \tag{33}$$

where

$$
\begin{cases}
\mu = r_N(\Gamma_{min}) \dfrac{\tau}{\Delta\Gamma} \left[ \dfrac{r_N(\Gamma_{min})^8}{9r_0^8}(\rho^9 - 1) - \dfrac{\Gamma_{min} + \tau}{\tau}(\rho - 1) \right] & \text{(34a)} \\[2ex]
\sigma = \left\{ r_N(\Gamma_{min})^2 \dfrac{\tau}{\Delta\Gamma} \left[ \dfrac{r_N(\Gamma_{min})^8}{45r_0^8}(\rho^{10} - 1) - \dfrac{2r_N(\Gamma_{min})^8}{9r_0^8}(\rho - 1) - \dfrac{\Gamma_{min} + \tau}{\tau}(\rho - 1)^2 \right] - \mu^2 \right\}^{1/2} & \text{(34b)} \\[2ex]
\text{and:} \\[1ex]
\rho = \dfrac{r_N(\Gamma_{max})}{r_N(\Gamma_{min})} & \text{(34c)}
\end{cases}
$$

where Eqn. 31 was substituted into Eqn. 29 and Eqns. 15 and 20 were used to calculate Eqns. 34a and 34b. Higher moments of the distribution are theoretically calculable by repeated integration by parts, from which additional facts may be learned about the distribution.

These calculations can be made analytically because of the ability to write the equation of the boundary and because that equation is invertible for the unknown parameter, that is, it can be written as $r_N(\Gamma)$ (a boundary in Euclidean space) and as $\Gamma(r_N)$ (a boundary in parameter space). This example also serves to highlight the asymmetry of the PHM despite the underlying uniform distribution of physical parameters. This asymmetry is highlighted by the significant spread in the mean, median, and modal boundaries (Fig. 2).

This example was greatly simplified by the assumption of radial symmetry, allowing the distribution to be cast onto one random variable, the radius of the flow boundary, giving integral curves of constant azimuthal angle which are linear in map view. However, in most realistic cases, this is a very poor assumption and the distribution will vary between integral curves.

## 3.2 Example 2: Mohr-Coulomb Flow on an Inclined Plane

To see the full realization of these concepts, a true two-dimensional problem is required. A simple example flow which had been well-studied is a lab-scale granular flow of sand down an inclined plane with a Mohr-Couloumb (MC) friction relation analogous to a natural-scale debris avalanche or landslide (Patra et al., 2005; Dalbey et al., 2008; Yu et al., 2009; Webb and Bursik, 2016; Aghakhani et al., 2016; Patra et al., 2018a, b). In the simulated version, a MC rheology variant of the Saint-Venant shallow water equations is solved numerically with the purpose-built solver TITAN2D (available from vhub.org). In this setup (Fig 3a), a $5$ cm-radius, $5$ cm-high cylindrical pile is placed $0.7$ m up a $1$ m long, $38.5°$ inclined plane and is allowed to flow out onto a flat runout $0.5$ m-long by $0.7$ m-wide. In accordance with MC theory, the flows were stopped locally when the sum of the drag due to internal and bed friction exceeded the gravitational forces and the simulations were stopped globally after $1.5$ s, when the flow dynamics had typically ceased (Patra et al., 2018b). In this case, the solution variable of interest $h$ is the maximum flow height over time, that is, the maximum flow height that each point experiences through the course of the flow.

The uncertain parameters in the problem for a given granular medium are the internal and basal friction angles of the medium, $\varphi_{int}$ and $\varphi_{bed}$ respectively. To construct the PHM for this flow, $512$ TITAN2D simulations were performed over a

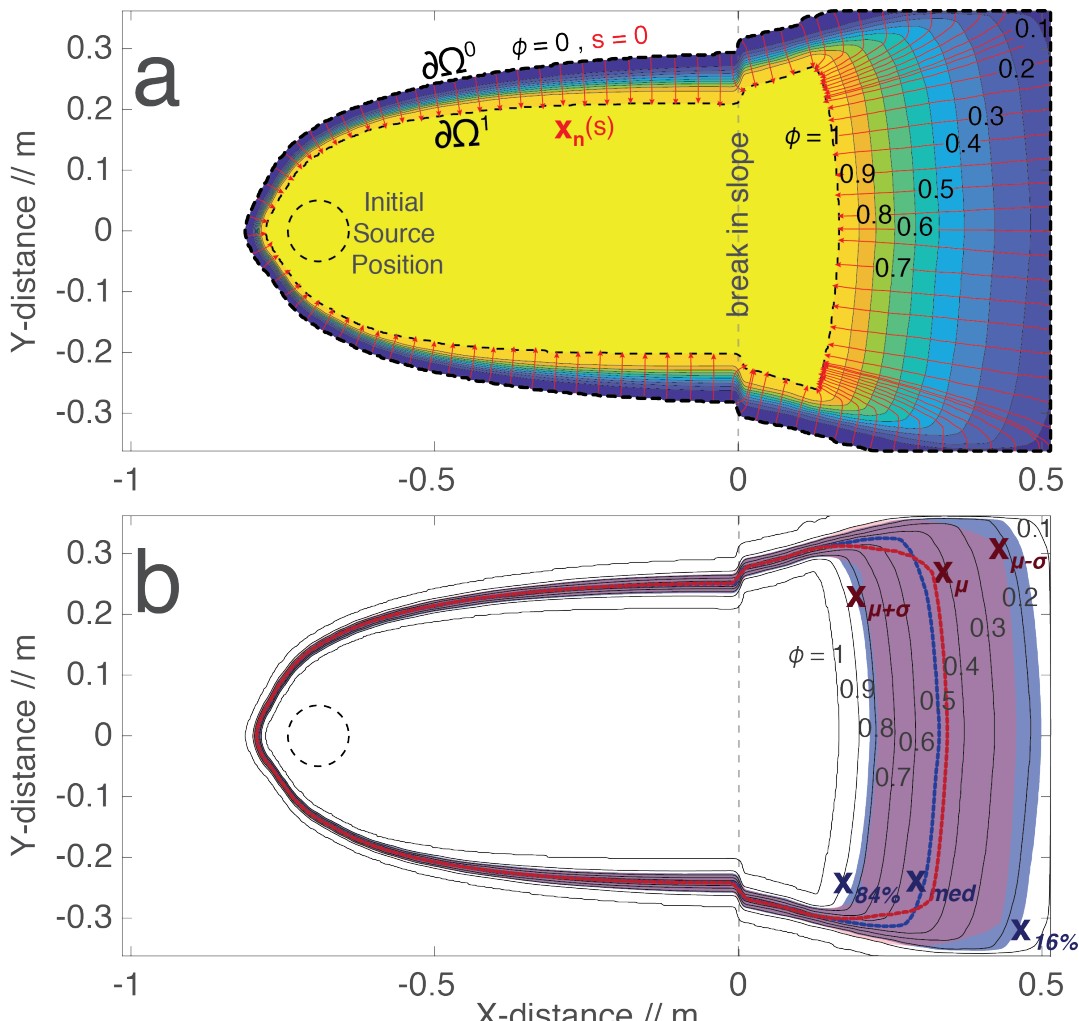

**Figure 3.** a) PHM contours for a Mohr-Coulomb (MC) flow on an inclined plane with integral curves $\boldsymbol{x_n}(s)$ (red) connecting points on $\partial\Omega^0$ to points on $\partial\Omega^1$ as defined in the text. b) Mean ($\boldsymbol{x}_\mu$, red curve) and median ($\boldsymbol{x}_{med}$, blue curve) flow boundary sets as well as regions within one standard deviation from the mean (red) and containing $68\%$ of the possible flow boundaries around the median (blue).

Latin-Hypercube-sampled input parameter space of

$$\mathcal{B} = \left\{ (\varphi_{int}, \varphi_{bed}) \,\middle|\, 18° \leq \varphi_{bed} \leq 30° \ \cap \ 2° \leq \varphi_{int} - \varphi_{bed} \leq 10° \right\}. \tag{35}$$

To construct the indicator functions for this experiment, a thickness threshold of $h_0 = 1.0$ mm was adopted. The PHM was then computed by averaging the indicator functions: a discrete analogy to Eqn. 4. Owing to the discrete nature of the data, all

5    continuous operations on the PHM were done by numerical finite differences and quadrature.

The PHM for this problem has many important characteristics, notably that the greatest dispersion in the probability contours occurs in the runout zone, owing principally to the uncertainty in $\varphi_{bed}$. Additionally, the spreading angle of the flow as a whole experiences a significant increase as the flow passes the break in slope (Fig. 3a). In order to calculate the mean flow boundary set and higher moments of this distribution, the gradient ascent curves of Eqn. 5 were integrated numerically from a finite set of closely spaced points around an approximation of $\partial \Omega^0$ defined as

$$\widehat{\partial \Omega^0} := \left\{ \boldsymbol{x}^j \mid \phi(\boldsymbol{x}) = 10^{-4} , \ j \in [1, J] \right\} \tag{36}$$

where $J = 2049$ is the number of such points generated by numerical contouring. Eqn. 5 was integrated by Euler's method with a 1 mm step forward in $s$ ($\Delta s_k = 1$ mm), thus fixing $||\boldsymbol{x_n}^j(s_{k+1}) - \boldsymbol{x_n}^j(s_k)|| = 1$ mm. This yielded discrete integral curves $\boldsymbol{x_n}^j(s_k)$ of varying lengths around the perimeter of the PHM (Fig. 3a). With each $\boldsymbol{x_n}^j(s_k)$ calculated, $\phi(s_k; l^j)$ was constructed and the parameter $\mu^j$ calculated on each curve, yielding an estimate of $\boldsymbol{x_\mu}(l)$ by interpolation of the discrete points $\boldsymbol{x_\mu}^j$ (Fig. 3b). The standard deviations $\sigma^j$ and the standard deviation space curves $\boldsymbol{x_{\mu \pm \sigma}}(l)$ were calculated by similar means (Fig. 3b). An important result of these calculations is that for almost every part of the runout zone, the mean flow boundary set $\boldsymbol{x_\mu}$ is dislocated from the median flow boundary set $\boldsymbol{x}_{med}$ ($\phi = \frac{1}{2}$) indicating qualitatively the general asymmetry of this distribution. Additionally the region bounded by the curves $\boldsymbol{x_{\mu \pm \sigma}}$ is remarkably different from the region which bounds the central $68\%$ of the data - the percent bounded by $\pm \sigma$ in the normal distribution (Fig. 3). All of this indicates that for PHM's of even simple realistic flows, the underlying statistics are not only non-normal and non-symmetric, they are variable along the boundary of the hazard.

### 3.3 Example 3: PHM and PHDM Construction From Simulations of the 1955 Debris Flow in Atenquique, Mexico

Following from the previous application of the PHM statistics to a simple flow, here we present a more advanced application: construction and analysis of a PHM for a more complex flow over natural topography. This example consists of numerical modelling detailed in Bevilacqua et al. (2019) of the 1955 volcaniclastic debris flow which destroyed the village of Atenquique at the base of Nevado de Colima, Mexico. On 16 October, 1955, at 10:45 am, residents of Atenquique experienced the arrival of an 8-9 m-high debris flow wave front that subsequently destroyed much of the town's buildings and infrastructure and caused more than 23 deaths (Ponce Segura, 1983; Saucedo et al., 2008). The debris flow was initiated by landsliding in the upper reaches of the Atenquique, Arroyo Seco, Los Plátanos and Tamazula fault ravines, with a minimum initial volume of $3.2 \times 10^6$ m$^3$ (Rupp, 2004; Saucedo et al., 2008). In the simulation of this flow, this volume was increased by $10\% - 50\%$ to give volumes of $V = 4.25 \pm 0.75 \times 10^6$ m$^3$ (Bevilacqua et al., 2019).

We analyze an ensemble of TITAN2D simulations of this flow that were generated by Bevilacqua et al. (2019) using a physically realistic rheology for a debris flow, the Voellmy-Salm (VS) rheology. This rheology is parameterized by a bed friction coefficient $\mu_{VS}$ and a velocity-dependent friction parameter $\xi$ which approximates turbulence-induced dissipation and is uncertain over several orders of magnitude. Consequently, $\xi$ is sampled logarithmically (Patra et al., 2018a, b). The flow was initiated from 5 source paraboloidal piles of material with unitary aspect ratios and volumes represented as constant fractions of the total flow volume which were proportional to the drainage size (Fig. 4a). The input parameter space $\mathcal{B}$ for this simulation

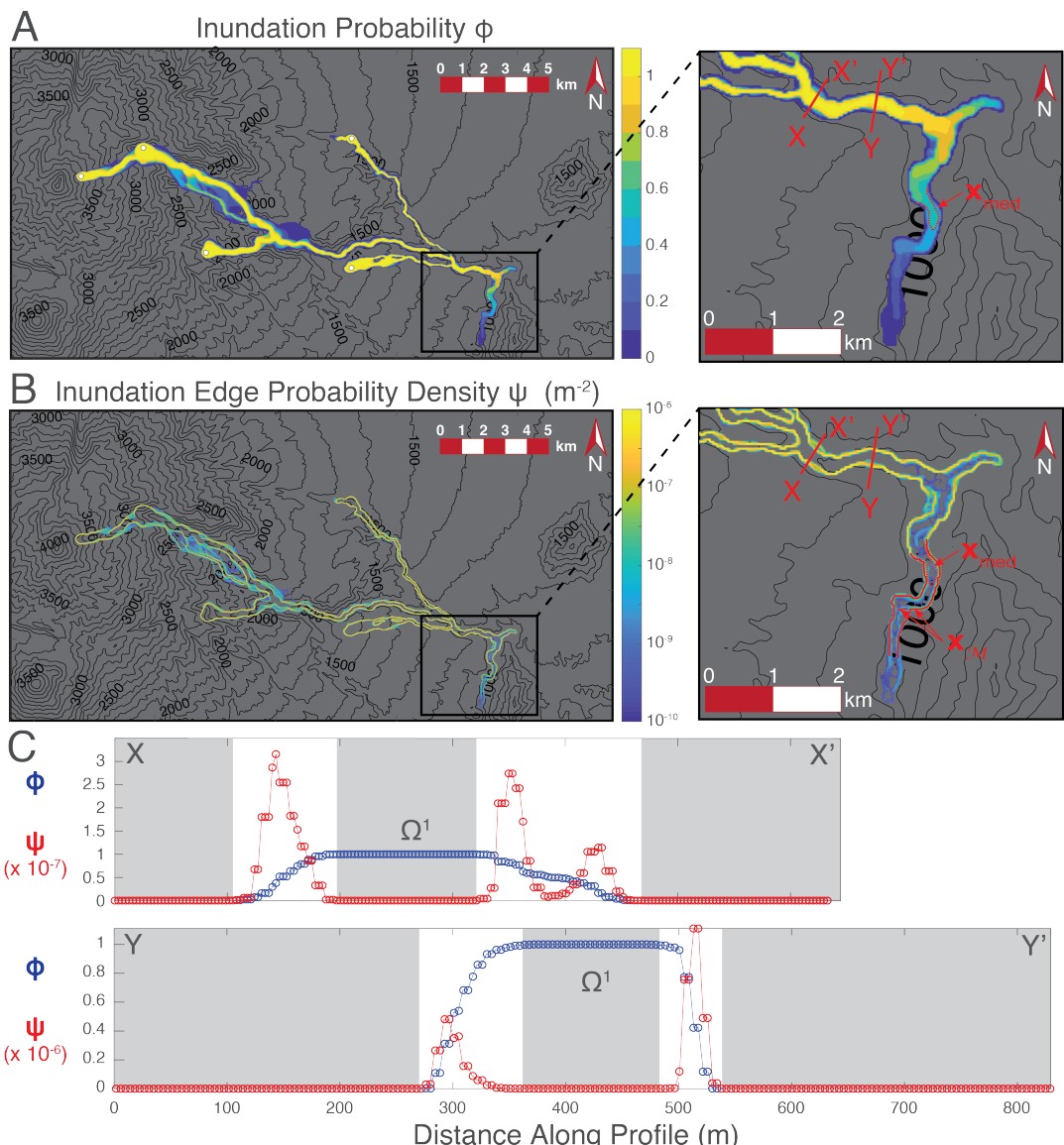

**Figure 4.** A) PHM and B) PHDM for a lahar inundating Atenquique, Mexico using the numerical model TITAN 2D incorporating the parameter space described in the text. C) Profiles of the PHM and PHDM across sections X-X' and Y-Y' highlighting the fact that each side of the PHM profile acts as a CDF for the location of the inundation edge with the corresponding scaled PDF given by the same profile along the PHDM. Grey background (C) indicates regions where $\phi = 0$ or $\phi = 1$ ($\psi = 0$).

was taken as a convex subset of $\{(\arctan(\mu_{VS}), \log_{10}(\xi), V)\}$ with the bounds $1° \leq \arctan(\mu_{VS}) \leq 3°$, $3 \leq \log_{10}(\xi) \leq 4$ and $3.5 \times 10^6 \leq V \leq 5.0 \times 10^6$. This set was determined using information about the known flow runout, flow arrival time in the

town, and exclusion of ravine overflow in conjunction with a round of preliminary simulations according to a new procedure of probabilistic modelling set-up (Bevilacqua et al., 2019).

As in the previous example, the solution variable of interest is the maximum flow height over time; however, here the height threshold $h_0 = 5.0$ cm is used. The 350 VS simulation indicator functions were averaged as in the previous example to obtain the PHM for this flow (Fig. 4a). For such a complex flow margin, the gradient-ascending integral curves could not be computed reliably at the resolution of the model; however, the PHDM was calculated using a discrete gradient operation, giving a sense of the maximum likelihood for the inundation boundary (Fig. 4b). Owing to the complexity of this flow and its underlying topography, profiles of the PHM and PHDM across the flow indicate that each margin of the flow is distributed non-symmetrically, although the sense of asymmetry does not appear to be systematic (Fig. 4c). In numerous locations, the flow boundary distributions (profiles of the PHDM) are multimodal which are interpreted to be controlled primarily by complexities in the underlying topography, relating to partial spill-over of the steep bounding ravines. Of particular interest is the behavior of the PHM and PHDM at the distal margin of the flow in the town of Atenquique and as it leaves town down river to the south and up river to the north. To the south (down river) The PHM decays over a very long distance ($> 5$ km) down a constrained channel, giving small values for the PHDM along the direction of flow (Fig. 4b). Consequently in this region, the flow boundary distributions are multimodal with weaker modes more distally and only the lateral boundaries being well-defined. This portion of the flow highlights some important problems with PHMs in general, they may include regions where the distribution spans a wide range of distances and will therefore have a large standard deviation, yielding high uncertainty in the flow boundary location. The accuracy of the PDFs over the integral curves in this region is most sensitive to the number of samples affecting this region. In a finite sampling of the parameter space, these PDFs may contain spurious peaks in regions where very few models reach. For example if only a small number of models reach a region where the PHM decays over many grid cells, unless some smoothing or interpolation is applied before gradient calculation, the PHM will decay step-wise with many grid spaces between steps. When the numerical gradient is calculated over a set cell size, these steps will become peaks in the PDFs and PHDM. This problem is generally absent where the PHM step size is more comparable to the scale of the grid spacing used by the gradient operator.

This complex example highlights the need for consideration of additional probability measures than simply the probability contours. Where the flow is not narrowly confined or where spill-over has occurred, the shape of the distribution becomes relevant and may yield significantly different results than estimating the likely flow boundary as one of the probability contours.

## 4  Discussion

As shown by the above three examples, our theory can be applied to give consistent results between continuous, analytic models and discrete, numerical models of flows. The following discussion highlights several consequences of the theory as applied to analytic models in research and more complex numerical models in hazard assessments.

## 4.1 Analytic Models

Although an analytic model (Quick et al., 2017) was invoked in the first example to make an exact calculation of the mean, median, mode, and standard deviation of the flow boundary, the real value of these models is that they are generally invertible and can be used to extract information about the input parameters from the properties of a given flow. Specifically, the PDF measuring the space of input parameters $f$ can be inverted from data collected from the natural phenomena that the analytic model represents. Using the example of the freezing viscous gravity current highlighted above (Quick et al., 2017), we show that an ensemble of natural examples of such flows can be used to invert the PDF of input parameters directly from the definition of the PHM. Suppose that remote sensing or geological field work yields data on the maximum radii ($r_N$) for an ensemble of these natural freezing gravity currents and that a natural PHM model is constructed called $\hat{\phi}(r)$ which represent the discrete CDF of radii that were measured. As before, we consider a simplified case wherein the most uncertain parameter from among the collection of natural flows is the freezing rate parameter $\Gamma$. If the researcher were sufficiently confident that these assumptions and the governing equation (Eqn. 21) accurately describe these flows, then the only differences between $\hat{\phi}(r)$ and Eqn. 29 would arise as a consequence of the PDF $f(\Gamma)$ of the natural input space being non-uniform. Consequently, because the formula for the flow boundary ($r_N(\Gamma) = r_0(1 + \Gamma/\tau)^{1/8}$) is known and is invertible, the formula for the PHM can also be inverted directly in this simple case as follows:

$$\hat{\phi}(r_N(\Gamma)) = \int_{\Gamma_{min}}^{\Gamma_{max}} f(\Gamma)\mathbf{1}_{\Omega(\Gamma)}(r_N(\Gamma))\,\mathrm{d}\Gamma = \int_{\Gamma(r_N)}^{\Gamma_{max}} f(\Gamma')\,\mathrm{d}\Gamma'. \tag{37}$$

Applying the Fundamental Theorem of Calculus and the chain rule yields

$$f(\Gamma) = -\frac{d}{d\Gamma}\hat{\phi}(r_N(\Gamma)) = -\frac{dr_N}{d\Gamma}(\Gamma) \cdot \frac{d\hat{\phi}}{dr_N}(r_N(\Gamma)). \tag{38}$$

In practice, the collected data would be in the form of a normalized histogram $\hat{p}(r_N)$ for the maximum flow radii. As a discrete approximation, $\hat{\phi}(r_N)$ is the associated complementary CDF for $\hat{p}$. Consequently, the PDF over parameter space could be written approximately as

$$f(\Gamma_i) = \frac{dr_N}{d\Gamma}(\Gamma_i) \cdot \hat{p}(r_N(\Gamma_i)) \tag{39}$$

where $\Gamma_i$ are sample points at which the calculation is performed. This illustrates the general fact that the distribution of flow boundaries observed reflects the underlying parameter space PDF weighted by the Jacobian (univariate derivative in this case) of the analytic mapping from parameter space to the flow boundary location in Euclidean space. Generally, unless the analytic model predicts that the boundary of a flow ($\Omega(\boldsymbol{\beta})$) is proportional to a given parameter, the input space distribution $f(\boldsymbol{\beta})$ undergoes a nonlinear transformation in the construction of a PHM and PHDM. More precisely, we remark that the calculated distribution is the conditional probability density $f(\Gamma|\beta_k)$, where $\beta_k$ represents all of the other parameters that were assumed to be known precisely. Inverting the PHM for the joint parameter space PDF is beyond the scope of this work.

## 4.2 Numerical Models and Probabilistic Hazard Assessment

If the parameter space of a given inundation hazard scenario and a realistic physical model of the process are known, a simple numerical experiment can be run using classical Monte Carlo methods or other Monte Carlo variants as was done for the granular flow on an inclined plane (Patra et al., 2018b) or the 1955 Atenquique lahar (Bevilacqua et al., 2019), as well as even more sophisticated sampling methods (e.g., Bursik et al, 2012; Bayarri et al., 2015). As in the examples in this study, the hazard map and hazard density map and the associated statistics may be generated from such experiments. This may then be used to give additional information to a probabilistic assessment of the hazard rather than merely the inundation probability contours. In this context, the method outlined here is capable of generating the typical measures that would appear in routine statistical analyses of distributions. The most obvious among these is the mean flow boundary and standard deviation region. These are fundamentally newly calculable measures for PHMs and should be included in probabilistic hazard assessments.

As illustrated in the examples, the mean boundary may differ significantly from the median boundary owing to the asymmetric statistics that are inherent to typical nonlinear flows. Furthermore, this asymmetry suggests that the modal or maximum likelihood boundary may be of interest to users of these assessments as well. For the scientific component of probabilistic hazard assessments to have maximum impact and accuracy, these analyses should incorporate the full statistics of the PHM.

## 5 Conclusions

With new tools to calculate the moments of any given hazard edge distribution from a PHM, hazard map makers and analysts become able to estimate the likely location of the hazard edge and the uncertainty in that estimate. As numerical modelling capabilities continue to grow, the increased ease and speed of this type of analysis will allow the scientific community to quickly construct the PHM, PHDM, the mean hazard boundary curve, standard deviation region, as well as higher moments of the hazard boundary distribution, giving improved estimates of the true hazard zone. Although this theory is useful for many applications where probabilities are given spatially, further work is needed to constrain which types of hazard assessments and models are most suitable for this analysis and how best to blend this scientific information within existing hazard map-making procedures. Similarly, additional work is required to study the probabilistic inverse problem further than the simple example given and to evaluate the use of this theory in a time - dependent context. Overall, the analysis put forth here significantly enhances the study of spatial probabilistic hazards, yielding new estimates of the likely hazard boundary and the uncertainty in those estimates.

## Appendix A:  PHDM Normalization

Using the definition of the integral curves in the text as steepest ascent integral curves $\boldsymbol{x_n}(s)$ such that

$$\frac{d\boldsymbol{x_n}}{ds} = \hat{\boldsymbol{n}} := \frac{\nabla\phi}{|\nabla\phi|} \tag{A1}$$

with $0 \leq s < L$ where $\boldsymbol{x_n}(0) \in \partial\Omega^0$ and $\boldsymbol{x_n}(L) \in \partial\Omega^1$, we prove the following lemma.

**Lemma: PHDM Normalization.** *Consider a probabilistic hazard map (PHM) $\phi(\boldsymbol{x})$ with the following properties (as given in the text):*

*(i) $\phi(\boldsymbol{x})$ has compact support in a region $\Omega^0 \subset \mathbb{R}^2$, with $\phi = 0$ on $\partial\Omega^0$*

*(ii) $0 \le \phi \le 1$*

*(iii) There exists a (possibly unconnected) set $\Omega^1 \subset \Omega^0$ in which $\phi(\boldsymbol{x}) = 1$*

*(iv) $\phi(\boldsymbol{x}) \in C^2(\mathbb{R}^2 \setminus \{\partial\Omega^0 \cup \partial\Omega^1\})$*

*(v) If $\phi$ has local maxima $\phi(\boldsymbol{x}_m) < 1$ with $m \in 1...M$ then each $\boldsymbol{x}_m$ can be surrounded by a region of influence $\mathcal{R}_m$ bounded by integral curves as defined in the text.*

*Define:*

$$\widetilde{\Omega}^0 := \Omega^0 \setminus \bigcup_{m=1}^{M} \mathcal{R}_m. \tag{A2}$$

*Then:*

$$\int_0^L |\nabla\phi(\boldsymbol{x_n}(s))|\,\mathrm{d}s = 1 \implies \int_{\widetilde{\Omega}^0} |\nabla\phi(\boldsymbol{x})|\,\mathrm{d}A = -\int_{\widetilde{\Omega}^0} \phi\nabla\cdot\hat{\boldsymbol{n}}\,\mathrm{d}A. \tag{A3}$$

*Proof.* Consider the level sets of $\phi(\boldsymbol{x})$ as integral curves everywhere tangent to the level sets, given by

$$\frac{d\boldsymbol{x_t}}{dl} = \mathbf{R}\hat{\boldsymbol{n}}, \tag{A4}$$

where $\mathbf{R} = \begin{pmatrix} 0 & -1 \\ 1 & 0 \end{pmatrix}$ is a rotation matrix. The curves $\boldsymbol{x_t}(l)$ are the level sets of $\phi$. By invoking the tangent curves and parameterizing each normal curve $\boldsymbol{x_n}(s;l)$, each PDF $\frac{d\phi}{ds}(s;l)$ may be extended away from a single integral curve:

$$\mathrm{d}l \int_0^L |\nabla\phi(\boldsymbol{x_n}(s;l))|\,\mathrm{d}s = \mathrm{d}l. \tag{A5}$$

Integration through $l$ yields an integral of the gradient magnitude over $\widetilde{\Omega}^0$ with the area element parameterized in local orthonormal curvilinear coordinates by $(s,l)$ which are derived from the PHM instead of ordinary space. Since both sets of integral curves $(\boldsymbol{x_n}, \boldsymbol{x_t})$ are parameterized by arc length, then $\mathrm{d}A(\boldsymbol{x}) = \mathrm{d}A(s,l)$. Because of these facts, the value of the integration will depend on the $\phi$ particular to the problem under consideration. Note that in the coordinates $(s,l)$, each of the domains of influence related to local maxima less than unity ($\mathcal{R}_m$) is a rectilinear patch of $(s,l)$ space (segment of $l \in [0,\infty)$) which is removed, leaving the integration domain of $l$ which contains gaps:

$$\mathcal{L} = \mathbb{R}_+ \setminus \bigcup_{m=1}^{M} \mathcal{R}_m(l). \tag{A6}$$

The right hand side of Eqn. A3 can then be evaluated by invoking a special case of the divergence theorem. Introducing an outward unit normal $\boldsymbol{\nu}$ on $\partial\widetilde{\Omega}^0$ and noting that:

$$\begin{cases} \phi = 0 \text{ on } \partial\Omega^0 \cap \partial\widetilde{\Omega}^0 & \text{(A7a)} \\[2mm] \hat{\boldsymbol{n}} \cdot \boldsymbol{\nu} = 0 \text{ on each segment } \partial\widetilde{\Omega}^0 \cap \partial\mathcal{R}_m & \text{(A7b)} \\[2mm] \partial\widetilde{\Omega}^0 = (\partial\Omega^0 \cap \partial\widetilde{\Omega}^0) \cup \bigcup_{m=1}^{M} (\partial\widetilde{\Omega}^0 \cap \partial\mathcal{R}_m), & \text{(A7c)} \end{cases}$$

the divergence theorem gives

$$\int_{\widetilde{\Omega}^0} \nabla \cdot (\phi\hat{\boldsymbol{n}})\,\mathrm{d}A = \int_{\widetilde{\Omega}^0} |\nabla\phi| + \phi\nabla \cdot \hat{\boldsymbol{n}}\,\mathrm{d}A = \int_{\partial\widetilde{\Omega}^0} \phi\hat{\boldsymbol{n}} \cdot \boldsymbol{\nu}\,\mathrm{d}l \equiv 0. \tag{A8}$$

From this result, integration of Eqn. A5 will yield:

$$\int_{\mathcal{L}} \int_0^L |\nabla\phi(\boldsymbol{x_n}(s;l))|\,\mathrm{d}s\,\mathrm{d}l = \int_{\widetilde{\Omega}^0} |\nabla\phi|\,\mathrm{d}A = -\int_{\widetilde{\Omega}^0} \phi\nabla \cdot \hat{\boldsymbol{n}}\,\mathrm{d}A = \text{constant} =: Q \tag{A9}$$

which is finite and positive due to the smoothness assumed in defining the hazard map $\phi(\boldsymbol{x})$ and the fact that $|\nabla\phi|$ is everywhere non-negative. In cases where the gradient of the PHM tends to infinity, the smoothness of the PHM must be modified in the sense of distributions, allowing derivatives and integrals on the PHM in the sense of distributions. This defines the PHDM normalization $Q$ and proves the lemma. $\qquad\square$

## Appendix B: Recipe for Generation and Analysis of the PHM

The following procedure is designed to simplify and condense the many steps involved in the above analysis. This "recipe" has been written as a blueprint for carrying out the analysis using a numerical model of the process in question. This procedure assumes that a numerical model of the process has already been selected.

1. Identify the collection of uncertain input parameters ($\boldsymbol{\beta}$) that are to be varied for the experiment as well as their ranges, correlation structure, and joint distribution. This may be done by the methods of (Bevilacqua et al., 2019) or other methods chosen for the particular application. This process generates the joint PDF $f(\boldsymbol{\beta})$ measuring the input space.

2. Generate random sample input vectors $\boldsymbol{\beta}$ distributed according to $f$ and run the numerical model on these inputs. For example, Latin hypercube sampling is a well-established procedure for defining pseudo-random designs of samples in $\mathbb{R}^n$ for multivariate uniform probability distributions (McKay et al., 1979; Stein, 1987; Owen, 1992). If all of the parameters are uncorrelated but non-uniform, then each random variable may be sampled independently according to its marginal distribution alone. If $f$ measures a correlated set of input random variables, then a sampling scheme must be used which respects the correlation structure and marginal distributions of the random variables. This sampling is

very straightforward for the multivariate normal distribution which has implementations in most scientific computing languages; however, other distributions require different techniques such as the NORTA (NORmal To Anything) process (Cairo and Nelson, 1997). The number of samples generated is directly related to the smoothness of the output PHM to be created.

3. Identify the threshold of interest for a given model observable and generate the indicator function for each model run according to the threshold. In practice, this can be realized as a sequence of grids, each of which is an element-wise indicator. In a MATLAB environment, this could be accomplished for each grid with the following syntax in which the indicator and the variable of interest are arrays of double-precision floating-point:

```
INDICATOR = double( VARIABLE > THRESHOLD_VALUE )
```

4. The PHM $\phi$ is obtained by performing an element-wise average of all of these indicator grids. The random sampling according to $f(\boldsymbol{\beta})$ ensures that the element-wise average represents the underlying distribution. Some smoothing of $\phi$ may be required for further processing to reduce the effects of finite input sampling.

5. Generate the level set unit normal vector field for $\phi$ using a numerical gradient operation $\boldsymbol{g} \approx \nabla \phi$ which can be implemented in most scientific computing language. Each component of $\boldsymbol{g}$ is a grid of the same size as $\phi$. Calculate the unit normal vector field as:

$$\hat{\boldsymbol{n}} = \begin{cases} \boldsymbol{g}/|\boldsymbol{g}| & \text{for } |\boldsymbol{g}| \neq 0 \\ 0 & \text{elsewhere} \end{cases} \tag{B1}$$

or in a MATLAB script:

```
[g_x , g_y] = gradient( PHI , dx , dy )
g_mag = ( g_x.^2 + g_y.^2 ).^0.5
n_x = g_x ./ g_mag
n_y = g_y ./ g_mag
n_x( isnan(n_x) ) = 0
n_y( isnan(n_y) ) = 0
```

6. Generate the gradient-ascending integral curves $\boldsymbol{x_n}$. This is done numerically by any gradient-ascent algorithm for solving

$$\boldsymbol{x_n}(s_{k+1}) = \boldsymbol{x_n}(s_k) + \Delta s_k \, \hat{\boldsymbol{n}}(\boldsymbol{x_n}(s_k)). \tag{B2}$$

$\hat{\boldsymbol{n}}(\boldsymbol{x_n}(s_k))$ may be computed for each step by interpolating the vector field $\hat{\boldsymbol{n}}$ at the point $\boldsymbol{x_n}(s_k)$. The initial position for each curve is given by the set of points output by a numerical contouring algorithm to find points very close to the

$\phi = 0$ level set. A large number output points should be specified in the contouring. In practice, a very small level set value is chosen for this contouring ($\phi = 10^{-4}$ in our example 2). In general, small values of $\Delta s_k$ give smoother curves. Iteration follows for each curve until $g(x_n(s_k)) \to 0$ with either (a) $\phi(x_n(s_k)) = 1$ or (b) $\phi(x_n(s_k)) < 1$. These two conditions ensure that either the integral curve ends on $\partial\Omega^1$ (condition a) or a local maximum $\phi(x_m)$ (condition b).

7. Generate the valid regions of the PHM by eliminating all regions in which integral curves terminate at local maxima. In practice, this could be accomplished by generating an indicator grid in which all array elements within a certain neighborhood of these curves are set to zero and all other elements set to 1.

8. Generate the PHDM using the gradient magnitude $|g|$ that was already calculated. For the normalizing constant $Q$, choose a numerical integration scheme (trapezoidal rule, Simpson's rule, Monte Carlo, or others) and compute the integral over the entire grid of the gradient magnitude filtered by the valid region indicator created in the last step. The PHDM is then given as $\psi = |g|/Q$. Any integration of $\psi$ must be computed with the same method.

9. To compute the mean on each curve, treat the sequence $\phi_k = \phi(x_n(s_k))$ as samples of a CDF over the sequence $s_k$, numerically approximate the formula

$$\mu = \int_0^L 1 - \phi(s) \mathrm{d}s \tag{B3}$$

with a suitable technique (trapezoidal rule, Simpson's rule, or others) and calculate the mean position on each curve as $x_\mu = x_n(\mu)$ by interpolation. The variance on each curve can be found similarly by numerical approximation of

$$\sigma^2 = \int_0^L 2s(1 - \phi(s)) \mathrm{d}s - \mu^2. \tag{B4}$$

Higher moments can be calculated by similar formulas involving the term $1 - \phi(s)$ which comes from integration by parts of the classical probability distribution moment formulas. The median position $x_{med}$ on each curve is the point at which $\phi = 1/2$ and the mode or maximum likelihood position $x_\mathcal{M}$ is at the maximum of $\psi(x_n(s))$, which can be found by interpolation of the PHDM along the computed curves. Each $x_\mu$, $x_{med}$, and $x_\mathcal{M}$ is then connected for all of the integral curves, making a set of curves for each of the mean, median and mode, and a region bounded by two curves for the standard deviation.

*Competing interests.* The authors declare no competing interests.

*Acknowledgements.* The present work was funded by NASA grant NNX12AQ10G, the JPSS PGRR program under NOAA-University of Wisconsin CIMSS Cooperative Agreement number NA15NES4320001, NSF awards 1339765, 1521855, and 1621853, and private donations to the University at Buffalo Foundation. Additionally, we thank the University at Buffalo Center for Computational Research.

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
