# Peer review of "Statistical theory of probabilistic hazard maps: a probability distribution for the hazard boundary location"

_Natural Hazards and Earth System Sciences, 2018_

## Referee Comment (RC1) · Anonymous Referee #1 · 16 Dec 2018

The paper introduces a formal probabilistic framework which may be used to provide additional scientific hazard information. I am very positive about these attempts to make the hazard calculation formally robust, and so I am also positive about a publication on NHESS. However, I think that the paper requires some modifications and/or clarifications that should be addressed in a revised version. My main suggestions follow (not in order of importance).

- I think that the authors should make an effort to simplify their terminology. As I said in my review to a previous paper of the same group of scientists, such a simplification could facilitate the reading of this paper to a wider audience of volcanologists. At the

same time, I think that a simplification in terminology can be made without loosing the scientific and mathematical rigor.

- At page 2 the authors mention that using categorical boundary in the hazard is very common. This statement should be articulated better. It is true what said by the authors and I would add that the use of categorical boundaries may facilitate the comprehension of the hazard map to laymen. However, it has been also highlighted by several authors the risk of this scientific discretization, in particular when the categories are associated to some decision making. If the discretization is made to facilitate the decision making, it cannot be made only through pure scientific arguments.

- At page 2 the authors claim "An important deficiency in the analysis of the PHM is that previously, estimates of the likely hazard boundary (a single curve on the map) have not been computed by consistent methods." In my opinion this statement is too strong, and should be justified. Personally, I know several papers which may a "consistent" hazard mapping. In my understanding, the method provided here can increase the scientific information related to the hazard mapping, but it does not demonstrate that 'all' previous PHM efforts are not based on consistent methods.

- The authors use the word "ensemble" as a collection of outcomes of one model with different initial and/or boundary conditions. Correctly at page 3 the authors say that their framework precludes the possibility to handle different models. They justify this choice claiming that the use of one model and an appropriate subspace of the general parameters are enough to explore the full variability. I think that this statement is too optimistic and should be modified. As a matter of fact, in many natural hazards different models are commonly used to estimate the so-called epistemic uncertainty. Even when the physics is very well known such as, for example, in predicting the space-time evolution of hurricanes, different models provide different paths (the physics is so complicated that some models focus on describing better some aspects instead of others). For this reason, the term "ensemble" has been generalized since the first applications to include also the variability among different models. I think that this

points has to be discussed in the manuscript.

- At page 3 the authors say that the main goal of their work is to build statistically meaningful boundary of the area impacted by a flow. In the paper they claim that this procedure can be applied to the full hazard or to conditional hazards (e.g., scenarios). I think that this method is much more meaningful for a conditional hazard than for the full hazard. In fact, the full hazard is usually the product of a combination of the outcomes of different scenarios. So, the probability represented by the contour lines often do not represent any specific scenario. In this case, it is not clear what is the meaning of calculating the boundary of the area impacted by a flow, because, for example, different realistic scenarios can be either smaller or larger than the average value (which may be not related to any possible scenario). Conversely, when this procedure is applied to a specific conditional hazard, i.e., to the hazard provided by one specific scenario, calculating the boundary of the area impacted by a flow makes much more physical sense.

- In essence, in most practical cases the method formally estimates a PDF from an empirical cumulative function (the PHI parameter in the manuscript) which may be estimated by counting the frequency of simulations for which one specific site is hit by a flow. I think it may be interesting to compare the method with a simple and straightforward numerical derivation of the cumulative. I am aware that most of the times the numerical derivative is quite noisy, but I am curious to see if it is the same for this case.

- At page 18, the authors write "For probabilistic hazard assessments to be used in sophisticated applications including risk assessments by governments or actuarial assessment for insurance purposes, the full statistics of the PHM must be considered.". I think that this statement is too dogmatic. It is not the role of scientists to say what is important for the risk, but just to provide a wide range of possible outcomes of the hazard analysis. This is exactly what this paper does, but I would not say that the decision makers "must" use this kind of information instead of others.

- In the conclusions the authors write "hazard edge location and the uncertainty in that estimate.". This statement is confusing to me. Are they talking about the aleatory variability or about the epistemic uncertainty? These two interpretations have a quite different physical interpretation.

---

## Referee Comment (RC2) · Anonymous Referee #2 · 22 Mar 2019

The authors present an interesting point of view regarding Probabilistic Hazard Maps (PHM) adopted in the volcanological context. PHM, typically, represent the probability of emplacement of volcanic products on the ground and are built by repeated application of models which explore a set of input parameters. The authors show that the PHM also represents a set of cumulative density function for the location of the inundation boundary. This allows the generation of probability density functions that can be used for further statistical analysis (mean, mode, median and other moments of the position of the inundation front). The authors, unavoidably, adopt a mathematical language often unusual among volcanologists, however I think that the use of examples clarify how the method can be applied.

[Figure]

I suggest to summarize the method by writing a sort of schematic "recipe", possibly applied to the numerical case, which is more often used in real cases (eg: something like, step_1: definition of the sampled input space parameters; step_2: definition of a threshold for the inundation; step_3: construction of the indicator function by using model simulations; step_4: definition of a set of starting point for the integration of the gradient curves [eq.(37) and eq.(5)], etc., up to the definition of the PDF).

---

## Author Comment (AC1) · 1 May 2019

**1   Overall notes on the revised manuscript:**

In our revised manuscript we have added the suggestions of the reviewers including descriptions of the uncertainty, clarifications of the purpose and role of the study in probabilistic hazard mapping, and inclusion of a "recipe" that simplifies the steps in carrying through the described processes.

Otherwise, we have made one moderate revision to our manuscript which was not suggested by the reviewers, namely, we have redefined (with normalization) the calculation

of the integral curves, thus slightly redefining the PHDM. Specifically, we have scaled the integral curve ordinary differential equations (ODE) by the unit- normalized gradient of the PHM, leading to integral curves parameterized by arc length. We have done this for a few reasons. There include:

1.) It is conceptually easier for the audience since the curve parameter measures Euclidean length along the curve.

2.) It propagates uncertainty in parameter space more directly to uncertainty in map (Euclidean) space.

3.) All integrals are now proper since the integration bounds become finite.

4.) Numerically, calculating the integral curves is easier since the ODEs involve a direction field and are insensitive to large gradients.

5.) Physically, the units are more logical, specifically, the arc length parameter, the mean and standard deviation have units of length, the curve-generating ODEs are dimensionless, and the PHDM has units of probability per length squared, consistent with a two-dimensional density function.

Overall, this constitutes a relatively small change to the mathematics; however, these changes have been propagated throughout the manuscript, including recalculation of the data in the figures. This change has not impacted the general procedure, the results (except in the model data presented), nor the discussion or conclusions.

**2   Responses to Comments of Reviewer 1**

"The paper introduces a formal probabilistic framework which may be used to provide additional scientific hazard information. I am very positive about these attempts to make the hazard calculation formally robust, and so I am also positive about a publica-

tion on NHESS. "

**Thank you.**

"However, I think that the paper requires some modifications and/or clarifications that should be addressed in a revised version. My main suggestions follow (not in order of importance)."

Comment 1

"I think that the authors should make an effort to simplify their terminology. As I said in my review to a previous paper of the same group of scientists, such a simplification could facilitate the reading of this paper to a wider audience of volcanologists. At the same time, I think that a simplification in terminology can be made without loosing the scientific and mathematical rigor."

**Thank you for your concern. We believe that the notation we use, including the set notation, is necessary. We recognize that the discussion of the local maxima introduces a great deal of notation; however, it is too important to move to an appendix and cannot be described precisely without it. We have made efforts throughout the exposition of our method in section 2 to explain the key notations and mathematical ideas wherever they appear. We have added extensive explanation at the beginning of the mathematical information to ensure that the reader is on firm footing before proceeding. At the recommendation of the second reviewer, we have included (in appendix B) the entire method as a simplified, conceptual "recipe," which we believe explains the method in looser, less technical terms.**

Comment 2

"At page 2 the authors mention that using categorical boundary in the hazard is very common. This statement should be articulated better. It is true what said by the authors and I would add that the use of categorical boundaries may facilitate the comprehension of the hazard map to laymen. However, it has been also highlighted by several authors the risk of this scientific discretization, in particular when the categories are associated to some decision making. If the discretization is made to facilitate the decision making, it cannot be made only through pure scientific arguments."

**Thank you for raising this point. We recognize that nuance is required for the successful interaction of scientific analysis and decision making, and so we have modified these statements to reflect that this procedure should only be considered as a formal scientific analysis and that in matters of map creation for wide distribution, a variety of other considerations may be considered in delineating hazard zones. This study should only be viewed as formalizing the scientific component of these considerations.**

Comment 3

"At page 2 the authors claim "An important deficiency in the analysis of the PHM is that previously, estimates of the likely hazard boundary (a single curve on the map) have not been computed by consistent methods." In my opinion this statement is too strong, and should be justified. Personally, I know several papers which may a "consistent" hazard mapping. In my understanding, the method provided here can increase the scientific information related to the hazard mapping, but it does not demonstrate that 'all' previous PHM efforts are not based on consistent methods."

**We have modified this statement to "An important deficiency in the analysis of the PHM is that previously, scientific estimates of the likely hazard boundary (a**

**single curve on the map) have lacked the formal framework required for unique determination of statistical estimates such as the mean, variance, and higher moments of the distribution of hazard boundary locations." We make this statement because the integral curves we introduce here are required to calculate these estimates uniquely, that is, the integral curves are the only way to profile the PHM so that the mean curve (for example) is dense in the sense of having the topology of a continuous curve or that of continuous curve segments in the case that local maxima are present.**

Comment 4

"The authors use the word "ensemble" as a collection of outcomes of one model with different initial and/or boundary conditions. Correctly at page 3 the authors say that their framework precludes the possibility to handle different models. They justify this choice claiming that the use of one model and an appropriate subspace of the general parameters are enough to explore the full variability. I think that this statement is too optimistic and should be modified. As a matter of fact, in many natural hazards different models are commonly used to estimate the so-called epistemic uncertainty. Even when the physics is very well known such as, for example, in predicting the space-time evolution of hurricanes, different models provide different paths (the physics is so complicated that some models focus on describing better some aspects instead of others). For this reason, the term "ensemble" has been generalized since the first applications to include also the variability among different models. I think that this points has to be discussed in the manuscript."

**We agree that our method is made maximally concrete by considering only a single model, that is a single function from inputs to outputs; however, many of the concepts in our method are applicable to PHM constructed from multiple models. We do not stipulate that our analysis precludes an ensemble of**

**models, rather we formulate our analysis in terms of a single model with an ensemble of inputs for conceptual simplicity for the benefit of the reader. Indeed, a multi-model approach can be cast in a specialized probability space where the ensemble of models characterizes epistemic uncertainty. Of course, this probability space will be measured by a probability density function, which may itself be epistemically uncertain; however, in this case, we can invoke the notion that such an analysis represents a single point in a time-dependent sequence of analyses subject to current knowledge. All of the probabilistic boundary estimation is invariant to this distinction, it only requires those mathematical properties of the PHM that we outline in the middle of section 2.1. We have included a discussion of aleatory and epistemic uncertainty in section 1 which makes the case that these types of analyses can include a mixture of aleatory and epistemic uncertainties as long as the epistemic uncertainties can be represented by a probability distribution. Because of this, any probabilistic hazard assessment made using this assumption could be considered as one in a time-dependent sequence of such assessments that would hopefully improve as epistemic uncertainty is reduced. These nuances are now stated in section 1.**

Comment 5

"At page 3 the authors say that the main goal of their work is to build statistically meaningful boundary of the area impacted by a flow. In the paper they claim that this procedure can be applied to the full hazard or to conditional hazards (e.g., scenarios). I think that this method is much more meaningful for a conditional hazard than for the full hazard. In fact, the full hazard is usually the product of a combination of the outcomes of different scenarios. So, the probability represented by the contour lines often do not represent any specific scenario. In this case, it is not clear what is the meaning of calculating the boundary of the area impacted by a flow, because, for example, different realistic scenarios can be either smaller or larger than the average value (which may

be not related to any possible scenario). Conversely, when this procedure is applied to a specific conditional hazard, i.e., to the hazard provided by one specific scenario, calculating the boundary of the area impacted by a flow makes much more physical sense."

**In general, we agree that this analysis is most conceptually concrete for a PHM of a particular hazard or scenario. However, formally this analysis is valid for any PHM subject to the mathematical conditions stated in the text. In the case of a PHM for the full hazard, a calculation of the boundary would simply represent the likely boundary of the full hazard. In a more formal discussion of the hazard probability space for a full hazard (modelling the probability of impact from any volcanic hazard at a particular volcano), the full hazard can be though of as the set of events constructed from the union of multiple elementary events (scenarios). In this way, the boundary-finding analysis has a concrete meaning even for multiple hazards. We stipulate that the probabilistic boundary analysis is separate from the particular modelling done to produce the PHM. Subject to the conditions on the function phi stated in the text, the boundary analysis is possible with no knowledge whatsoever of the underlying model used to construct the PHM. Consequently, a PHM describing the full hazard is as able to be analyzed as a PHM for a single scenario.**

Comment 6

"In essence, in most practical cases the method formally estimates a PDF from an empirical cumulative function (the PHI parameter in the manuscript) which may be estimated by counting the frequency of simulations for which one specific site is hit by a flow. I think it may be interesting to compare the method with a simple and straight-forward numerical derivation of the cumulative. I am aware that most of the times the numerical derivative is quite noisy, but I am curious to see if it is the same for this case."

[Figure]

**Our "recipe" in appendix B is tailored to the specifics of performing our method on a numerical example, including the explanation of numerical derivatives and integration. We note here in response to this comment that both our example 2 and example 3 do what the reviewer requests, that is, perform the method with a numerical derivative of the cumulative distribution (phi). The value of the numerical gradient along the computed curves is the same as the numerical derivative (finite difference) of the sequence of phi_k with respect to the sequence of s_k.**

**2.1 Comment 7**

"At page 18, the authors write "For probabilistic hazard assessments to be used in sophisticated applications including risk assessments by governments or actuarial assessment for insurance purposes, the full statistics of the PHM must be considered.". I think that this statement is too dogmatic. It is not the role of scientists to say what is important for the risk, but just to provide a wide range of possible outcomes of the hazard analysis. This is exactly what this paper does, but I would not say that the decision makers "must" use this kind of information instead of others."

**Thank you for pointing this out. We seek to establish these methods as the standard mathematical methods to quantify probabilistic hazards. We have modified this statement to read "For the scientific component of probabilistic hazard assessments to have maximum impact and accuracy, these analyses should incorporate the full statistics of the PHM."**

**Comment 8**

"In the conclusions the authors write "hazard edge location and the uncertainty in that estimate.". This statement is confusing to me. Are they talking about the aleatory variability or about the epistemic uncertainty? These two interpretations have a quite

different physical interpretation."

**As we mentioned above, we have included a brief discussion of aleatory and epistemic uncertainty germane to this process. We wish to convey the notion that probabilistic hazard assessments of this type can be viewed as individual members of a time-dependent sequence of such assessments, each of which captures the types of uncertainty encoded in the generation of the PHM that is understood at that time.**

**3 Responses to Comments of Reviewer 2**

"The authors present an interesting point of view regarding Probabilistic Hazard Maps (PHM) adopted in the volcanological context. PHM, typically, represent the probability of emplacement of volcanic products on the ground and are built by repeated application of models which explore a set of input parameters. The authors show that the PHM also represents a set of cumulative density function for the location of the inundation boundary. This allows the generation of probability density functions that can be used for further statistical analysis (mean, mode, median and other moments of the position of the inundation front). The authors, unavoidably, adopt a mathematical language often unusual among volcanologists, however I think that the use of examples clarify how the method can be applied."

**Thank you.**

Comment 1

"I suggest to summarize the method by writing a sort of schematic "recipe", possibly applied to the numerical case, which is more often used in real cases (eg: something like, step_1: definition of the sampled input space parameters; step_2: definition of

a threshold for the inundation; step_3: construction of the indicator function by using model simulations; step_4: definition of a set of starting point for the integration of the gradient curves [eq.(37) and eq.(5)], etc., up to the definition of the PDF)."

**Thank you for this suggestion. We have included a "recipe" as you describe as the appendix B. This recipe carries through the various steps we formalize in the text with explicit details on practical calculations including sample MATLAB routines for some of the constructions.**

Minor and grammatical changes made to the manuscript are not presented here. This paper has not been published elsewhere, and represents new and original work. My co-authors have approved the manuscript and agree with its submission to *NHESS*.

Best Regards,

David Hyman, PhD

Cooperative Institute for Meteorological Satellite Studies (CIMSS)
Space Science and Engineering Center
University of Wisconsin - Madison
1225 W. Dayton St., Room 219
Madison, WI 53706 USA
Email: dave.hyman@ssec.wisc.edu
or dhyman2@wisc.edu

---

## Author Comment (AC3) · 1 May 2019

Attached is the revised manuscript for the article "Statistical theory of probabilistic hazard maps: a probability distribution for the hazard boundary location" by David Hyman, Andrea Bevilacqua, and Marcus Bursik.

The revisions undertaken are those detailed in the interactive discussion.

Please also note the supplement to this comment:
https://www.nat-hazards-earth-syst-sci-discuss.net/nhess-2018-344/nhess-2018-344-AC3-supplement.pdf

[Figure]

**Supplement:**

[revised manuscript text omitted]